

# Primary and secondary organic aerosol from heated cooking oil emissions

Tengyu Liu[1], Zhaoyi Wang[2,3], Xinming Wang[2,4,*], and Chak K. Chan[1,5*]

1. School of Energy and Environment, City University of Hong Kong, Hong Kong, China

2. State Key Laboratory of Organic Geochemistry and Guangdong Key Laboratory of Environmental Protection and Resources Utilization, Guangzhou Institute of Geochemistry, Chinese Academy of Sciences, Guangzhou, China

3. University of Chinese Academy of Sciences, Beijing, China

4. Center for Excellence in Urban Atmospheric Environment, Institute of Urban Environment, Chinese Academy of Sciences, Xiamen, China

5. City University of Hong Kong Shenzhen Research Institute, Shenzhen, China

*Corresponding author:

Chak K. Chan

School of Energy and Environment, City University of Hong Kong

Tel: +852-34425593; Fax: +852-34420688

Email: Chak.K.Chan@cityu.edu.hk

Xinming Wang

State Key Laboratory of Organic Geochemistry

Guangzhou Institute of Geochemistry, Chinese Academy of Sciences

Tel: +86-20-85290180; Fax: +86-20-85290706

Email: wangxm@gig.ac.cn



**Abstract**
Cooking emissions have been identified as a source of both primary organic aerosol
(POA) and secondary organic aerosol (SOA). To examine the characteristics of SOA
from cooking emissions, emissions from seven vegetable oils (sunflower, olive, peanut,
corn, canola (rapeseed), soybean, and palm oils) heated at 200 °C were photooxidized
under high-$NO_x$ conditions in a smog chamber. OA was characterized using a high-
resolution time-of-flight aerosol mass spectrometer (HR-TOF-AMS). Sunflower,
peanut, corn, canola, and soybean oil generated relatively low concentrations of POA
($\leqslant$ 0.5 µg m$^{-3}$) in the chamber. For palm and olive oil, positive matrix factorization
(PMF) analysis separated POA and SOA better than the residual spectrum method.
Temporal trends in concentrations of POA from heated palm oil were accurately
predicted assuming first-order POA wall loss. However, this assumption overestimated
POA concentrations from heated olive oil, which was attributed to the heterogeneous
oxidation of POA. The mass spectra of the PMF resolved POA factor for palm oil, and
the average POA from sunflower, peanut, corn, and canola oils were in better agreement
($\theta$ = 8–12°) with ambient cooking organic aerosol (COA) factors resolved in select
Chinese megacities than those found in given European cities in the literature. The mass
spectra of SOA formed from heated cooking oils had high abundances of $m/z$s 27, 28,
29, 39, 41, 44, and 55 and displayed limited similarity ($\theta$ > 20°) with ambient semi-
volatile oxygenated OA (SV-OOA) factors. The entire OA data set measured herein
follows a linear trend with a slope of approximately -0.4 in the Van Krevelen diagram,
which may indicate oxidation mechanisms involving the addition of both carboxylic



acid and alcohol/peroxide functional groups without fragmentation and/or the addition
of carboxylic acid functional groups with fragmentation.





## 1   Introduction


Organic aerosol (OA) contributes greatly to atmospheric particulate matter (PM)
(Kanakidou et al., 2005), which influences air quality, climate, and human health
(Hallquist et al., 2009). OA commonly comprises primary organic aerosol (POA)
emitted directly from sources and secondary organic aerosol (SOA) formed via the
oxidation of organic gases (Donahue et al., 2009). Cooking is an important source of
both POA (Abdullahi et al., 2013) and SOA (Liu et al., 2018). In aerosol mass
spectrometer (AMS, Aerodyne Research Incorporated, USA) measurements, cooking
OA (COA) has been found to contribute 10–35% of OA in urban areas (Allan et al.,
2010; Sun et al., 2011, 2012; Ge et al., 2012; Mohr et al., 2012; Crippa et al., 2013a, b;
Xu et al., 2014; Lee et al., 2015), although the AMS may overestimate COA due to the
COA relative ionization efficiency (1.56–3.06), which is higher than the typical value
of 1.4 used for organics (Reyes-Villegas et al., 2018). In particular, Lee et al. (2015)
found that the average contribution of COA to OA (35%) was even higher than that of
traffic-related hydrocarbon-like OA (HOA, 26%) at a roadside site in Mongkok in Hong
Kong. Xu et al. (2014) also observed higher contributions of COA (24%) than HOA
(16%) to OA in Lanzhou, China.

Although most of the COA mass spectra resolved by positive matrix factorization

(PMF) analysis (Paatero, 1997; Paatero and Tapper, 1994; Ulbrich et al., 2009; Zhang
et al., 2011) in the wider AMS dataset have the same basic characteristics, including
predominant peaks at $m/z$s 41, 43, 55, and 57 and high $m/z$ 55/57 ratios, the specific
COA mass spectra vary among studies (Mohr et al., 2009; Allan et al., 2010; Sun et al.,





2011; 2012; Ge et al., 2012; Mohr et al., 2012; Crippa et al., 2013a, b; Lee et al., 2015;
Elser et al., 2016; Struchmeier et al., 2016; Aijala et al., 2017). Differing cooking styles
may be among the factors that induce this variability in COA mass spectra. For instance,
the fraction of *m/z* 41 was higher than that of *m/z* 43 in COA mass spectra for Chinese
cooking (He et al., 2010), while the reverse was found for meat cooking (Mohr et al.,
2009). Atmospheric aging may also diversify the COA mass spectra. Significantly
different COA mass spectra have been resolved during summer and winter in Greece
despite the fact that cooking activities are similar during the two seasons (Florou et al.,
2017; Kaltsonoudis et al., 2017). In addition, the COA factors resolved by PMF analysis
may include emissions from other sources (Dall'Osto et al., 2015) and sometimes
cannot be separated from the other factors (Kostenidou et al., 2015; Qin et al., 2017).
The multilinear engine (ME-2) is a relatively newly developed tool that can use mass
spectra input from the literature to constrain the OA source apportionment solutions
(Canonaco et al., 2013). Qin et al. (2017) found that inputting different COA profiles
resulted in proportions of COA (to total OA) that differed by factors of up to 2.
Comparing laboratory-generated COA mass spectra with ambient PMF factors can help
to improve COA source apportionment. Previous studies have been focused on
investigating mass spectra of POA from cooking (Mohr et al., 2009; Allan et al., 2010;
He et al., 2010; Reyes-Villegas et al., 2018); however, studies exploring mass spectra
of SOA from cooking remain scarce.

Recent smog chamber studies have demonstrated that cooking emissions can form

large amounts of SOA via photochemical aging (Kaltsonoudis et al., 2017; Liu et al.,




2017, 2018). Kaltsonoudis et al. (2017) observed similarities between aged COA mass
spectra from meat charbroiling and corresponding POA mass spectra. Liu et al. (2017)
also reported extensive similarities ($R^2$ of 0.83–0.96) between mass spectra of POA and
SOA from heated cooking oils. These mass spectral similarities make it difficult to
separate POA and SOA from cooking in smog chamber experiments in terms of both
abundance and mass spectral signatures.

PMF has been used widely to deconvolve ambient AMS datasets, but relatively

less to analyze smog chamber data. Presto et al. (2014) used PMF to analyze POA and
SOA from vehicle exhaust. Kaltsonoudis et al. (2017) performed PMF analysis on fresh
and aged OA from meat charbroiling. However, PMF analysis has not been applied to
POA and SOA from heated cooking oils.
Sage et al. (2008) developed a residual spectrum method to separate POA and SOA
in diesel exhaust; this residual spectrum method assumes that all signal at $m/z$s 57 and
44 is associated with POA and SOA, respectively. Chirico et al. (2010) and Miracolo et
al. (2010) further improved the residual method, using the reduced $C_4H_9^+$ ion as a POA
tracer. Chirico et al. (2010) suggested that the appropriate POA tracer ion may differ
for different sources. The optimal tracer ions for cooking emissions remain unknown.
This study aims to characterize POA and SOA from heated cooking oils emissions,
obtain POA and SOA mass spectra via PMF analysis, and compare the resolved mass
spectra with those for ambient COA-related factors from PMF. We will also explore the
heterogeneous oxidation of POA from palm and olive oils.
**2   Materials and methods**



### 2.1 Smog chamber experiments


Seven photochemical aging experiments were conducted in a 30 m$^3$ indoor smog
chamber at the Guangzhou Institute of Geochemistry, Chinese Academy of Sciences
(Wang et al., 2014; Liu et al., 2015, 2016; Deng et al., 2017) (Table 1). The cooking
oils tested include sunflower, olive, peanut, corn, canola (rapeseed), soybean, and palm
oils, which together constitute over 90% of the vegetable oil consumed globally (USDA,
2017). All experiments were conducted at 25 °C and a relative humidity (RH) of less
than 5%. Prior to each experiment, the chamber was continuously flushed with purified
dry air for at least 48 h. The experimental procedures have been described in detail
elsewhere (Liu et al., 2018). Briefly, ammonium sulfate seed particles were introduced
first into the chamber to serve as condensation sinks to reduce organic vapor wall losses
(Zhang et al., 2014). Then, emissions from heated vegetable oils were introduced into
the chamber for 1–1.5 h by an air stream through a 2 m heated (70 °C) Teflon tube. The
emissions were generated by heating 250 mL of the target oil at approximately 200 °C
in a 500 mL flask in a dimethyl silicone oil bath. Nitrous acid (HONO) was then
introduced into the chamber as a source of hydroxyl radical (OH). The initial ratio of
non-methane organic gases (NMOGs) to NO$_x$ (NMOG:NO$_x$) fell largely between 2.6
and 5.4 ppbC:ppb, except for the palm oil experiment, in which it was 18.9 ppbC:ppb.
These ratios were larger than the typical urban ratio of ~3 (Gordon et al., 2014). After
the primary emissions had been characterized for at least 1 h, photochemical aging was
initiated by exposing the emissions to black lights (60 W Philips/10R BL365, Royal
Dutch Philips Electronics Ltd., the Netherlands) for 2–4.5 h.



Gas monitors were used to measure the concentrations of $NO_x$ and $O_3$ (EC9810,
9841T, Ecotech, Australia). NMOGs were characterized using a commercial proton-
transfer-reaction time-of-flight mass spectrometer (PTR-TOF-MS, Model 2000, $H_3O^+$
reagent ion, Ionicon Analytik GmbH, Austria) (Lindinger et al., 1998; Jordan et al.,
2009). Detailed descriptions of operating conditions, calibrations, and fragmentation
corrections can be found elsewhere (Liu et al., 2018). The decay of acrolein or
heptadienal was used to determine the OH concentration in the chamber.
A scanning mobility particle sizer (SMPS, TSI Incorporated, USA, classifier
model 3080, CPC model 3775) was used to measure particle number concentrations
and size distributions. The chemical composition of submicron non-refractory
particulate matter (NR-PM$_1$) was characterized using a high-resolution time-of-flight
aerosol mass spectrometer (hereafter AMS, Aerodyne Research Incorporated, USA)
(DeCarlo et al., 2006). The instrument alternated between the high-sensitivity V-mode
and the high-resolution W-mode every 1 min. The Squirrel 1.57I and Pika 1.16I toolkits
were used in IGOR (Wavemetrics Inc., USA) to analyze the AMS data; the Aiken et al.
(2008) fragmentation table was adopted. Elemental ratios, such as the hydrogen-to-
carbon ratio (H:C) and oxygen-to-carbon ratio (O:C), were determined using the
improved-ambient method (Canagaratna et al., 2015). HEPA-filtered particle-free air
from the chamber was measured for at least 20 min before and after each experiment to
determine the major gas signals. The ionization efficiency was calibrated using 300 nm
ammonium nitrate particles.
**2.2 Separating POA and SOA**



The experiments in this study were classified into two groups (Table 1). Sunflower,
peanut, corn, canola, and soybean oil emissions produced low POA concentrations (<
0.5 µg m$^{-3}$) in the smog chamber. Due to wall losses, the POA concentration was close
to the AMS detection limit when the lights were turned on (e.g., the sunflower oil
experiment in Supplementary Fig. S1). These experiments were therefore assumed to
involve only SOA. POA and SOA mixtures were present in the palm and olive oil
experiments, which produced maximum POA concentrations of 14 and 39 µg m$^{-3}$,
respectively. PMF analysis (Paatero, 1997; Paatero and Tapper, 1994) was performed
on the high-resolution mass spectra (*m/z*s 12–160) to deconvolve the POA and SOA
factors following the procedure of Ulbrich et al. (2009). PMF solutions were examined
for 1 to 5 factors with fPeak values varying from -1 to 1. Diagnostic plots are shown
for all datasets in Figs. S2 and S3. After examining the PMF residuals, time series for
different numbers of factors, and mass spectral similarity between PMF POA and
observed POA spectra, 2-factor solutions with fPeak values of -0.2 and 0 were chosen
for the palm and olive oil experiments, respectively.

The residual spectrum method (Sage et al., 2008; Chirico et al., 2010; Miracolo et

al., 2010; Presto et al., 2014) was used in addition to PMF analysis to separate the POA
and SOA. The residual method assumes that all tracer ion signal (e.g., C$_4$H$_9^+$) is
associated with POA and that the chemical composition of POA remains constant
throughout the entire experiment. The mass concentration of POA at time *t* can then be
calculated using the following equation:
$$POA_t = Ion_t / Ion_{t_0} \times OM_{t_0}, \qquad (1)$$



where $OM_{t0}$ is the total organic matter concentration at time $t = 0$, and $Ion_t$ and $Ion_{t0}$ are
the organic mass signals of a specific POA tracer ion at time $t$ and time $t = 0$ (lights on),
respectively. $C_4H_9^+$ is typically chosen as the POA tracer for combustion sources (Sage
et al., 2008; Chirico et al., 2010; Miracolo et al., 2010; Presto et al., 2014); the optimal
tracer ion for cooking emissions remain unclear. The tracer ions tested in this study
include $C_4H_7^+$, $C_4H_9^+$, $C_5H_8^+$, $C_5H_9^+$, $C_6H_9^+$, and $C_7H_9^+$.

## 3    Results and discussion

### 3.1 POA-SOA split

Figure 1 shows measured OA time series for the palm and olive oil experiments; the
OA concentrations were not corrected for particle wall loss. The experiments typically
involved introduction of the cooking emissions to the chamber (~1–1.5 h),
characterization of primary emissions (~1–2 h), and photochemical aging (2–4.5 h). In
the palm oil experiment, the POA concentration increased rapidly during the first ~1.6
h of the emission introduction period, reaching approximately 14 µg m$^{-3}$. The POA
concentration then decreased to ~9 µg m$^{-3}$ due to wall losses. SOA was quickly formed
after photochemical aging was initiated at $t = 0$, and the OA concentration increased by
a factor of 5 in less than 1 h. The maximum POA concentration in the olive oil
experiment was approximately two times higher than that in the palm oil experiment,
and the maximum OA concentration was ~50% of that in the palm oil experiment via
SOA formation. Palm oil produced SOA more efficiently than did olive oil; this is
consistent with our previous study, which found that the SOA production rate of palm
oil was 4 times that of olive oil, likely due to the higher abundance of SOA precursors





in palm oil emissions (Liu et al., 2018).
Figure 1 also shows time series of measured OA, resolved PMF factors, and POA
concentrations assuming first-order loss of POA to the walls. Two factors, namely POA
and SOA, were identified. Solutions with three or more factors introduced physically
inexplicable factors and did not improve the PMF performance (Fig. S4). Overall, the
OA reconstructed by PMF accurately captured the trends in measured OA throughout
the experiment. The sum of the residual was generally less than 2 µg m$^{-3}$, resulting in
ratios of total residual concentration ($\Sigma$Residual) to total OA concentration ($\Sigma$OA) of
less than 5% (Figs. S2 and S3). The separation of POA and SOA factors was reasonable
and interpretable. During the introduction and characterization of cooking emissions ($t$
< 0), the concentrations of SOA should be, by definition, exactly zero. The
concentrations of the resolved SOA factors were approximately 0.3 µg m$^{-3}$, capturing
the expected behavior.
In previous smog chamber studies of dilute emissions from combustion sources
(Weitkamp et al., 2007; Gordon et al., 2014; Liu et al., 2015, 2016), POA was typically
assumed to be inert, and POA concentrations followed first-order wall loss equations.
Figure 1 shows POA concentrations after the onset of photooxidation assuming first-
order wall loss. The wall loss rate constants were determined from the decay of POA
during the primary emissions characterization period. For palm oil, the predicted POA
concentrations agreed well with the PMF POA factor, suggesting that the assumption
of POA inertness was reasonable. However, for olive oil, the concentration of the PMF
POA factor was significantly lower after the onset of photooxidation ($t$ = 0) than the



POA concentration predicted assuming first-order wall losses. The PMF POA factor
decreased rapidly in the first hour after $t = 0$, which may have been due to the
heterogeneous oxidation of POA. Kaltsonoudis et al. (2016) observed similar changes
in POA concentrations when emissions from meat charbroiling were exposed to OH
levels similar to those used in this experiment. Nah et al. (2013) also observed rapid
heterogeneous OH oxidation of select cooking POA components such as oleic acid and
linoleic acid. The differences in POA behavior may have arisen from the different
chemical compositions, and corresponding differences in reactivity, of POA from olive
and palm oils.

To validate the differences in heterogeneous oxidation reactivity between POA

from palm and olive oils, two additional ozonolysis experiments were conducted
separately using an oxidation flow reactor. Emissions generated in the flask were first
passed continuously through the reactor for at least 30 min and then exposed to 500–
600 ppb of ozone ($O_3$) for another 17 min. The total flow rate and residence time in the
flow reactor were 6 L min$^{-1}$ and 75 s, respectively. OH radicals were not present in these
experiments. Previous work has demonstrated that exposing gas-phase emissions from
heated cooking oils to $O_3$ does not lead to SOA formation (Liu et al., 2017). Therefore,
any changes in OA chemical composition during these ozonolysis experiments were
attributed to heterogeneous oxidation. Figure 2 shows time series of O:C ratios and $O_3$
concentrations during the ozonolysis experiments. The olive oil O:C ratio increased
from 0.11 to 0.17 after the emissions were exposed to $O_3$ for 17 min; no obvious
changes were observed in the palm oil O:C ratio. These results demonstrate that POA



from olive oil undergoes heterogeneous ozone oxidation more readily than POA from
palm oil. The olive oil POA may contain more abundant unsaturated organic species,
which are expected to react quickly with OH radicals (Atkinson and Arey, 2003).
Figure 3 shows POA concentrations obtained using the residual method with
different POA tracer ions, along with the PMF-derived POA factors. For the palm oil
experiment, the residual method overestimated the POA concentrations (compared with
PMF) using all of the different tracer ions. For the olive oil experiment, the residual
method accurately predicted the POA concentrations before the lights were turned on
using different tracer ions. However, after photochemical aging began, the residual
method did not agree with the PMF results. Using $C_5H_9^+$ as a POA tracer, the residual
method captured the changes in POA during the first 1 h, but overestimated the POA
concentrations by as much as a factor of 2 for the remainder of the experiment; use of
the other ions led to significant POA concentration overestimation in comparison with
the PMF-resolved POA concentrations. The POA concentrations determined using
$C_4H_7^+$, $C_4H_9^+$, and $C_7H_9^+$ were higher even than those estimated for first-order POA
loss; this may be attributed to the presence of these ions in SOA, which is then
incorrectly allocated to POA by the residual method. These observations indicate that
the use of associated tracer ions to calculate POA is not valid for the photochemical
aging of cooking oil emissions.
Overall, PMF analysis effectively separated POA and SOA from heated cooking
oils. The traditional method, which assumes first-order POA wall loss, worked well
when the POA was inert, as in the palm oil experiment, but greatly overestimated the





POA concentration in the olive oil experiment, which was attributed to the occurrence
of heterogeneous oxidation. The residual method failed to capture the POA
concentrations using any of the different POA tracer ions due to the presence of these
ions in the SOA mass spectrum.
**3.2 Mass spectra of PMF-resolved factors**
Figure 4 shows mass spectra of POA emissions and PMF-derived POA factors from the
olive and palm oils. These were also compared to the average mass spectrum of POA
emitted from heated sunflower, peanut, corn, and canola oils obtained from Liu et al.
(2017). Overall, for both oils, the POA factor mass spectra agreed very well with the
directly measured POA spectra. For both oil types, the $\theta$ angles between the factor and
measured mass spectra were less than 5°; generally, $\theta$ angles of 0–5°, 5–10°, 10–15°,
15–30°, and > 30° indicate excellent agreement, good agreement, many similarities,
limited similarities, and poor agreement, respectively, between two mass spectra
(Kostenidou et al., 2009; Kaltsonoudis et al., 2017). The PMF analysis slightly
underestimated the mass fraction at $m/z$ 28 and slightly overestimated the mass fractions
at $m/z$s 41 and 55 in both experiments.

The olive oil POA factor was dominated by $m/z$ 41 ($f_{41} = 0.105$), followed by $m/z$s

69 ($f_{69} = 0.088$), 55 ($f_{55} = 0.075$), and 43 ($f_{43} = 0.050$). In the high-resolution mass
spectra, the most abundant ions in these unit masses were $C_3H_5^+$, $C_5H_9^+$, $C_4H_7^+$, and
$C_3H_7^+$, respectively (Fig. S5). The olive oil POA factor mass spectrum showed limited
similarity ($\theta = 26°$) to the average POA mass spectrum for the other cooking oils. The
$m/z$ 69 abundance in the POA factor was significantly higher than those (average $f_{69} =$





0.026) in POA from other oils, while the mass fractions of $m/z$s 29, 43, and 55 were
generally lower. The mass spectrum of olive oil POA measured directly in this study
also exhibited poor agreement ($\theta = 31°$) with olive oil POA mass spectra measured in
an oxidation flow reactor (Liu et al., 2017) (Fig. S6). The mass spectral differences
between this study and Liu et al. (2017) may have arisen from the different oil heating
temperatures and dilution conditions.

The palm oil POA factor was dominated by $m/z$ 55 ($f_{55} = 0.092$), followed by $m/z$s

41 ($f_{41} = 0.089$) and 43 ($f_{43} = 0.069$); the most abundant ions in these unit masses were
$C_4H_7^+$, $C_3H_5^+$, and $C_3H_7^+$, respectively (Fig. S5). The high abundances of $m/z$s 41 and
55 are similar to previous studies showing POA emissions from heated peanut, canola,
and sunflower oils (Allan et al., 2010), Chinese cooking (He et al., 2010), and meat
charbroiling (Kaltsonoudis et al., 2017). The POA factor mass spectrum exhibited good
agreement ($\theta = 9°$) with the average mass spectrum of POA from other cooking oils,
although the POA factor had relatively higher mass fractions of $m/z$s 67 and 69 and
lower abundances of $m/z$s 28 and 29. The palm oil POA factor had higher abundances
of oxygen-containing ions such as $CO^+$, $CHO^+$, $CO_2^+$, and $C_3H_3O^+$ than did the olive
oil POA factor, resulting in a relatively higher O:C ratio (palm O:C = 0.15; olive O:C
= 0.09).

Figure 5 shows PMF-derived SOA factor mass spectra of palm and olive oil and

the average mass spectrum of SOA formed from sunflower, peanut, corn, canola, and
soybean oils, which was obtained over a 10 min period after the OA concentration
reached its maximum for each oil. The SOA factors were dominated by $m/z$s 27, 28, 29,





41, 43, 44, and 55; the most abundant ions in these unit masses were $C_2H_3^+$, $CO^+$, $CHO^+$,
$C_3H_5^+$, $C_2H_3O^+$, $CO_2^+$, and $C_4H_7^+$, respectively (Fig. S5). The abundances of oxygen-
containing ions were generally higher than those in the corresponding POA factors. The
mass fraction at *m/z* 43 ($f_{43}$ = 0.087) was higher than $f_{44}$ (0.059) in the olive oil SOA
factor, while $f_{44}$ (0.074) dominated $f_{43}$ (0.067) in the palm oil SOA factor. Despite these
differences, the mass spectra of the two SOA factors exhibited good agreement ($\theta$ = 8°).
Although the olive oil and palm oil had different NMOG compositions (Liu et al., 2018)
and POA mass spectra, the SOA produced by the two oils had highly similar mass
spectra. The SOA factor mass spectra were also highly similar ($\theta$ = 15° for olive oil, 7°
for palm oil) to the average mass spectrum of SOA from five other cooking oils, which
contained a higher mass fraction of *m/z* 44 (average $f_{44}$ = 0.080). $C_4H_7^+$, $C_4H_9^+$, $C_5H_8^+$,
$C_5H_9^+$, $C_6H_9^+$, and $C_7H_9^+$ ions were present in both SOA factors, which led to the
incorrect separation of the POA and SOA by the residual method, as discussed above.
**3.3 Comparison of PMF-resolved factors with ambient factors**
Comparisons of mass spectra from laboratory-generated cooking OA with ambient
PMF-resolved COA factors can help to constrain COA source apportionment. Figure
6a summarizes $\theta$ angles between ambient COA factors from other studies and the palm
oil PMF POA spectrum, average PMF SOA spectrum, and average POA and SOA
spectra from cooking oils. The olive oil POA mass spectrum was not included in the
comparison as it can vary greatly under different experimental conditions. The
reference mass spectra from other studies were obtained from the AMS spectral
database (Ulbrich, I.M., Handschy, A., Lechner, M., and Jimenez, J.L., High-Resolution



333 AMS Spectral Database. URL: http://cires.colorado.edu/jimenez-group/HRAMSsd/)

334 (Ulbrich et al., 2009). The agreement between ambient COA factors and the palm PMF

335 POA and average POA mass spectra decreased from 8–12° for ambient COA factors

336 from the commercial and shopping area of Mongkok in Hong Kong (Lee et al., 2015)

337 to 25–28° for ambient COA factors from a suburban area in Pasadena (Hayes et al.,

338 2013). Cooking style may be among the factors that drives this variability in $\theta$. Mass

339 spectral agreement was better for urban areas in Hong Kong, Beijing, and Xi'an, where

340 stir-frying foods with vegetable oils is popular, than in urban European cities, where

341 grilling and broiling are prevalent (Abdullahi et al., 2013). Atmospheric oxidation may

342 also influence the correlations between the POA factor mass spectra found herein and

343 the ambient COA factors. For example, the mass spectrum of a cooking-influenced

344 organic aerosol (CIOA) factor identified in Pasadena (Hayes et al., 2013) displayed

345 better agreement with the SOA factor and average SOA mass spectra than with the POA

346 factor and average POA spectra. Kaltsonoudis et al. (2017) also found that this CIOA

347 factor correlated well with mass spectra of aged OA from meat charbroiling. The

348 average PMF SOA factor and average SOA spectrum derived herein were poorly

349 correlated ($\theta$ generally > 30°) with other ambient COA factors (Mohr et al., 2009;

350 Crippa et al., 2013a, b; Lee et al., 2015; Elser et al., 2016; Struchmeier et al., 2016;

351 Aijala et al., 2017). Our results suggest that one should consider the cooking style and

352 atmospheric oxidation conditions when constraining COA factors with the default COA

353 mass spectral inputs.





Figure 7 compares the average PMF SOA factor mass spectrum and average
ambient semi-volatile oxygenated organic aerosol (SV-OOA) factor (Ng et al., 2011a).
SV-OOA is ubiquitous in the atmosphere and generally associated with SOA (Ng et al.,
2010). The SV-OOA factor mass spectrum was recalculated following the Aiken et al.
(2008) fragmentation table, assuming that the OA mass at $m/z$ 28 was equal to that at
$m/z$ 44 and that $m/z$ 18 was equal to 22.5% of $m/z$ 44. The average mass spectrum of
the PMF SOA factors exhibited poor agreement with the SV-OOA average mass
spectrum ($\theta = 25°$). The average PMF SOA factor for cooking oils had higher
abundances of $m/z$s 27, 28, 29, 39, 41, 44, and 55 than did the SV-OOA factor and lower
mass fractions of $m/z$s 15 and 43. In particular, the SV-OOA spectrum had no signal at
$m/z$ 39, while the PMF SOA factor $f_{39}$ was 0.048. The average mass spectra of the PMF
SOA factors and SOA from other cooking oils were also unlike other ambient SV-OOA
factors ($\theta > 20°$) (Mohr et al., 2009; Crippa et al., 2013a; Hayes et al., 2013;
Struchmeier et al., 2016; Aijala et al., 2017) (Fig. 6b). The poor correlations between
cooking SOA and SV-OOA were not unexpected, as ambient SV-OOA may contain a
mixture of SOA from numerous sources, such as vehicle exhaust, biomass burning, and
industrial and biogenic emissions.
**3.4 OA oxidation state and chemical evolution**
Figure 8 shows the H:C and O:C ratios of PMF-resolved POA and SOA factors, SOA
from heated cooking oils, and ambient COA and SV-OOA factors in a Van Krevelen
diagram. The O:C ratio and estimated average carbon oxidation state ($OS_C$) ($OS_C \approx$
$218\times$ O:C $-$ H:C) (Kroll et al., 2011) generally increase with increasing atmospheric





OA aging. The O:C ratios for the olive and palm PMF POA were 0.09 and 0.15,
respectively, comparable to those found for Chinese cooking (0.08–0.13) and meat
charbroiling (0.10) (Kaltsonoudis et al., 2017). The O:C ratios for the olive and palm
PMF SOA and SOA from other cooking oils ranged from 0.40 to 0.50, slightly lower
than that of SV-OOA (0.53), which indicates that the SOA formed from cooking oils
herein was less oxidized than ambient SV-OOA. In Van Krevelen space, the entire
dataset features a linear trend with a slope of approximately -0.4, which may indicate
oxidation mechanisms involving the addition of both carboxylic acid and
alcohol/peroxide functional groups without fragmentation and/or the addition of
carboxylic acid functional groups with fragmentation (Heald et al., 2010; Ng et al.,
2011b). This slope is consistent with the aging of ambient OOA (Ng et al., 2011b) and
lower than the -0.8 slope noted in evolving ambient OA data corrected using the
improved-ambient method (Heald et al., 2010). The O:C ratios of these ambient COA
factors ranged from 0.11 to 0.34, consistent with the oxidation trends determined for
cooking OA in this study. Some of the COA factors had O:C ratios higher than those
noted for POA from cooking emissions in laboratory studies (e.g., O:C = 0.27 for CIOA
in Pasadena (Hayes et al., 2013) and O:C = 0.34 for COA in Atlanta (Xu et al., 2018));
it is possible that these COA factors contained aged OA and/or SOA formed from
cooking emissions.
**4   Conclusions**
SOA formation from heated cooking oil emissions was investigated in a smog chamber
under high-NO$_x$ conditions. For experiments with mixtures of POA and SOA, the POA





and SOA factors were separated using PMF, the traditional method, and the residual
method; PMF outperformed the other techniques in resolving accurate POA and SOA
factors. Although the traditional method, which assumes first-order POA wall losses,
worked well when the POA was inert, it greatly overestimated the POA concentration
when heterogeneous oxidation occurred. The residual method, which uses different ions
as POA tracers, failed to capture the POA concentrations due to the presence of these
ions in the SOA mass spectrum.

Mass spectra of palm oil PMF POA and average POA from other cooking oils

exhibited good agreement ($\theta$ = 8–14°) with ambient COA factors resolved in select
Chinese megacities such as Hong Kong, Beijing, and Xi'an and less similarity ($\theta$ = 11–
23°) with ambient COA factors determined in select European cities. The mass
spectrum of a CIOA factor determined in Pasadena was more consistent with the
average PMF SOA factor mass spectrum ($\theta$ = 17°) than with the POA factors ($\theta$ = 25–
28°). Our results suggest that one should consider the cooking style and atmospheric
oxidation conditions when performing deconvolution analyses with the default COA
mass spectral inputs.

The average mass spectra of PMF SOA factors and SOA from other cooking oils

exhibited little similarity ($\theta$ > 20°) to ambient SV-OOA factors, which is not unexpected
given that SV-OOA may contain a mixture of SOA from many sources. In the Van
Krevelen diagram, the entire data set in this study yielded a linear trend with a slope of
approximately -0.4, which may indicate oxidation mechanisms involving the addition





of both carboxylic acid and alcohol/peroxide functional groups without fragmentation
and/or the addition of carboxylic acid functional groups with fragmentation.





**Acknowledgments**
Chak K. Chan would like to acknowledge the support of the National Natural Science
Foundation of China (Project No. 41675117). Xinming Wang would like to
acknowledge the support of the Strategic Priority Research Program of the Chinese
Academy of Sciences (Grant No. XDB05010200).





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



**Table 1.** Experimental conditions in the photochemical aging experiments.

| Cooking oil | Dilution ratio | [NMOG]:[NO$_x$] (ppbC:ppb) | OH exposure molec cm$^{-3}$ s | Category | PMF |
|---|---|---|---|---|---|
| sunflower | 63 | 4.9 | $1.0 \times 10^{10}$ | Pure SOA | NA[a] |
| olive | 107 | 4.0 | $1.3 \times 10^{10}$ | POA+SOA | 2 factors |
| peanut | 67 | 2.6 | $2.1 \times 10^{10}$ | Pure SOA | NA |
| corn | 67 | 3.2 | $1.8 \times 10^{10}$ | Pure SOA | NA |
| canola | 67 | 5.4 | $3.5 \times 10^{10}$ | Pure SOA | NA |
| soybean | 67 | 3.4 | $1.7 \times 10^{10}$ | Pure SOA | NA |
| palm | 100 | 18.9 | $1.3 \times 10^{10}$ | POA+SOA | 2 factors |

[a] not applicable.





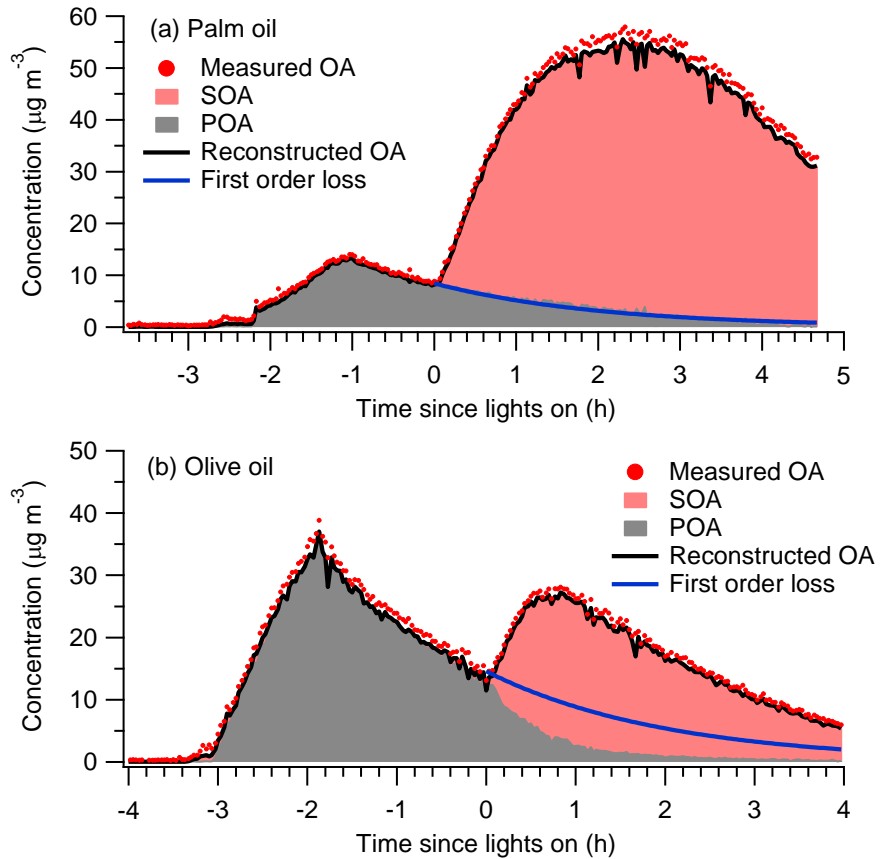

**Figure 1.** Time series of measured OA, PMF factors, and POA concentrations assuming

first-order loss of POA to the walls for (a) palm and (b) olive oil experiments.

Concentrations were not corrected for particle wall loss.

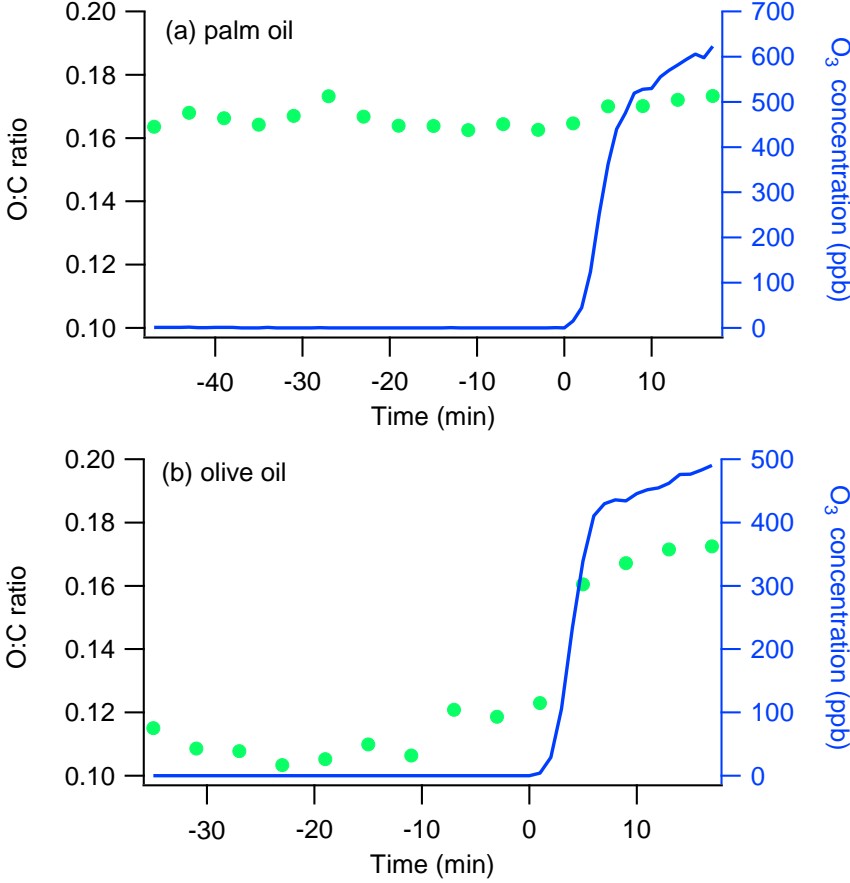

683

**Figure 2.** Time series of O:C ratios and O₃ concentrations for oxidation flow reactor

ozonolysis experiments with (a) palm and (b) olive oil. The ozonolysis experiments

involved exposing emissions from palm or olive oil to high concentrations of O₃ in an

oxidation flow reactor. The emissions first passed continuously through the reactor for

at least 30 min and then were exposed to 500–600 ppb ozone for another 17 min. Ozone

was introduced at time $t = 0$.



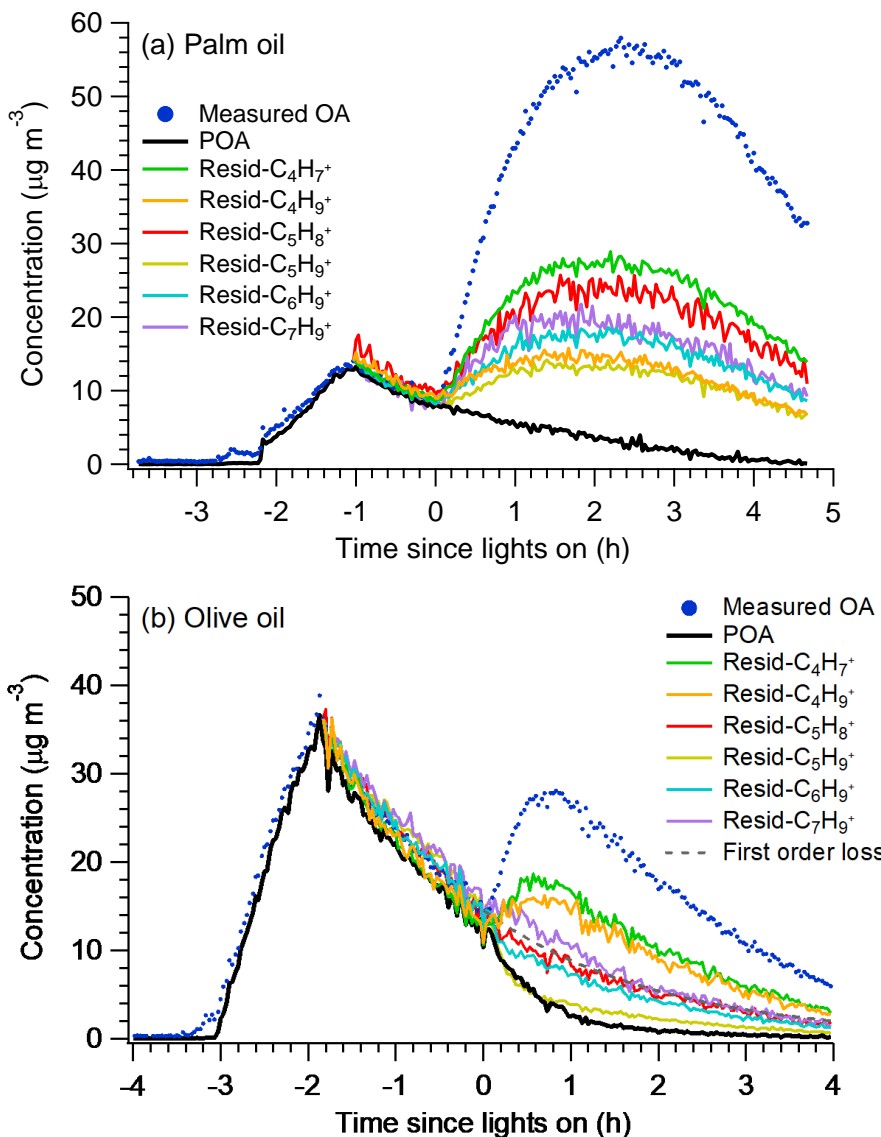


**Figure 3.** Time series of measured OA, PMF-derived POA factors, and POA

determined using the residual method for (a) palm and (b) olive oil experiments.






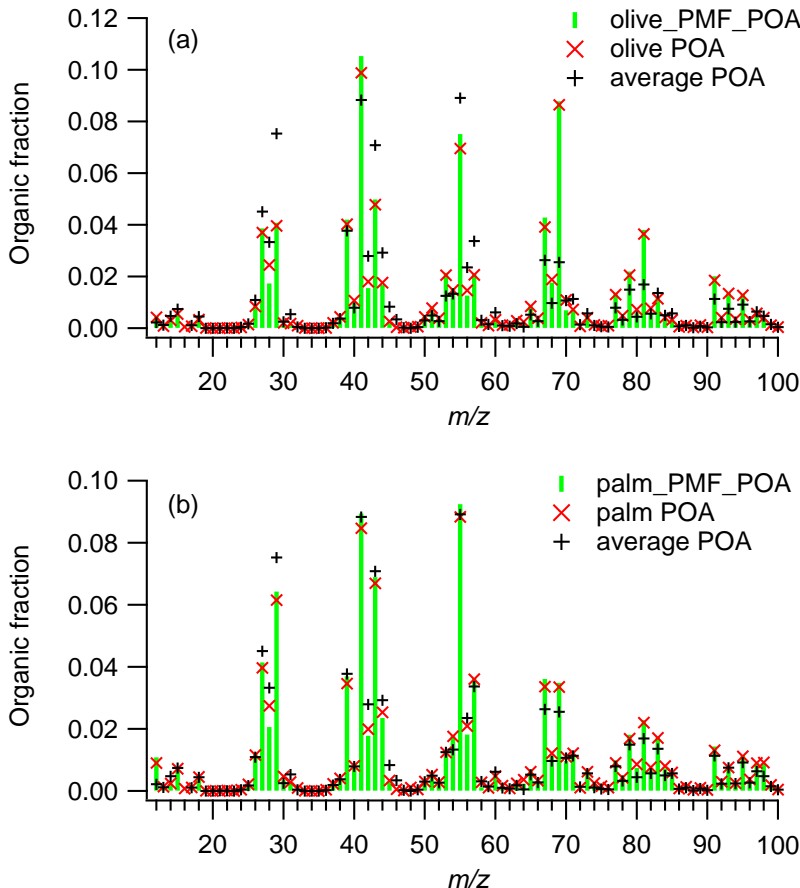


**Figure 4.** Mass spectra of POA emissions and PMF-derived POA factors for (a) olive

and (b) palm oil and, for comparison, the average mass spectrum of POA emissions

from sunflower, peanut, corn, and canola oils obtained from Liu et al. (2017).




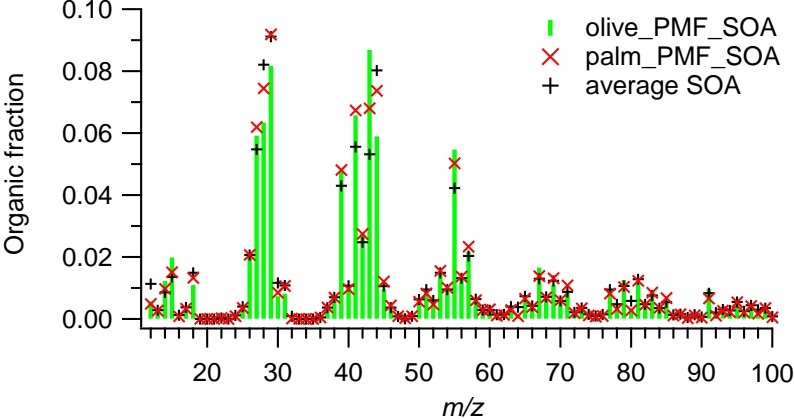


**Figure 5.** Mass spectra of PMF-derived SOA factors for (a) olive and (b) palm oil and,

for comparison, the average mass spectrum of SOA formed from sunflower, peanut,

corn, canola, and soybean oils. The SOA mass spectra were averaged over a 10 min

period after the OA concentration reached its maximum.





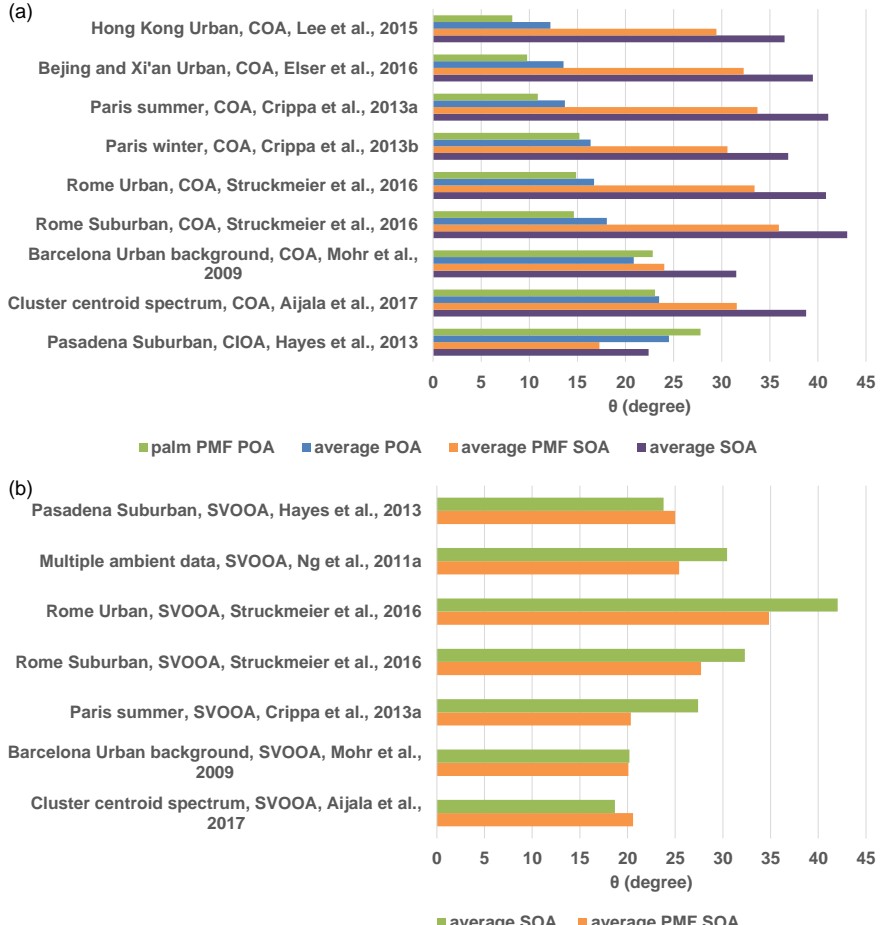

**Figure 6.** (a) Angles ($\theta$) between ambient COA factor mass spectra and the palm PMF

POA mass spectrum, average POA mass spectrum, average PMF SOA factor mass

spectrum, and average SOA mass spectrum. (b) Angles ($\theta$) between ambient SVOOA

factor mass spectra and the average PMF-derived SOA factor and average SOA mass

spectra. The average POA mass spectra were averaged for sunflower, peanut, corn, and

canola oils (Liu et al., 2017). The average SOA mass spectra were averaged for

sunflower, peanut, corn, canola, and soybean oils in this study.





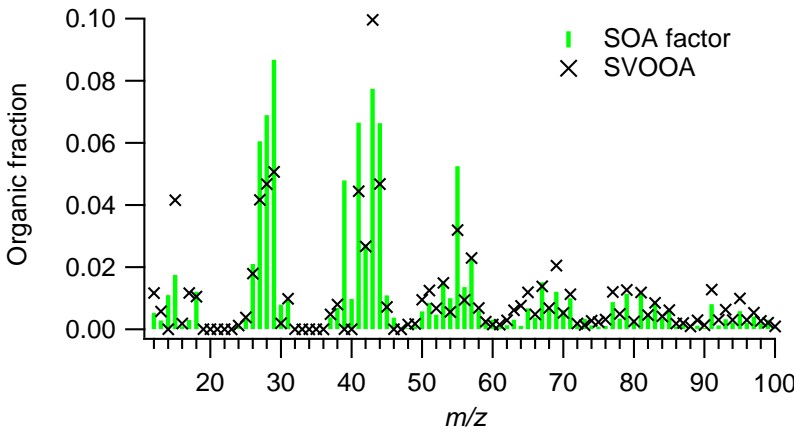


**Figure 7.** Average PMF SOA factor mass spectrum and ambient SV-OOA factor mass
spectrum (Ng et al., 2011a).






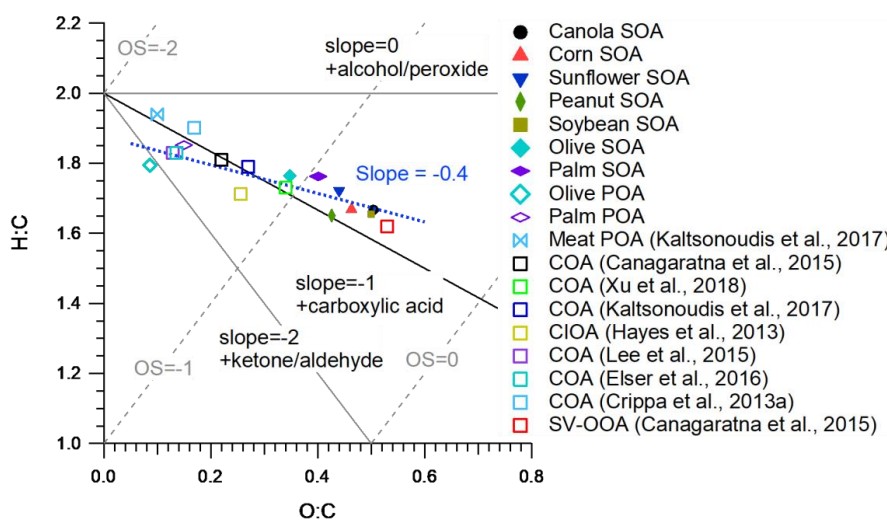


**Figure 8.** Van Krevelen diagram of POA and SOA from different cooking oils, as well

as ambient PMF factors. Average carbon oxidation states from Kroll et al. (2011) and

functionalization slopes from Heald et al. (2010) are shown for reference.