# Peer review of "Primary and secondary organic aerosol from heated cooking"

_Atmospheric Chemistry and Physics, 2018_

## Referee Comment (RC1) · Anonymous Referee #1 · 2 Jul 2018

Generally, this manuscript is well-written and contain good scientific content. I would like to recommend for publication if the authors could address some comments as the following: 1. Line 142-144. The author mention about the use of SMPS, but I have seen any results from these instrument. It is maybe better if the authors could show the evolution of particle number size distribution during the experiment. 2. Smog chamber experiments: I think the author should describe in more details about the experiment: How many experiments the authors conducted for each oil cooking? The time resolution of each instruments? 3. What about the uncertainty of the the number H:C, O:C for SOA that the author presented in the Fig.8?

---

## Referee Comment (RC2) · Anonymous Referee #2 · 16 Jul 2018

It has been well recognized that cooking emissions substantially contribute to atmospheric organic aerosols (OA), but studies on their characterization are scarce. Hence, investigation on primary and secondary OA contributions from such emissions is highly needed. This study focused on the characterization of POA and SOA produced from heated cooking oils in laboratory experiments, based on the obtained mass spectra via PMF analysis. The residual spectrum method has also been applied to separate the POA and SOA using different tracer ion signals. The authors have successfully separated the POA and SOA produced from heated palm and olive oil by PMF analysis. Further they found that residual spectrum method is not valid to calculate POA for photochemical aging of cooking oils. The results obtained from this study are interesting and the paper is written very well. Therefore, this paper is worthy to publish in ACP,

after addressing the following specific comments:

1. It has been noted (in Section 2.1; lines 142-144) that particle number concentrations and size distributions were measured using SMPS, but did not presented the obtained data in the paper. It is important to show the particle number concentrations and size distributions during the total period of the experiments, particularly during the period before the start of photochemical aging to understand the impact of gases absorption on to seed particles on the POA loading. 2. Surprisingly, the concentrations of POA emitted from heated palm oil and olive oil were gradually reduced starting from ∼1 h and 2 h, respectively, before the start of photochemical aging and reached to almost negligible level in 4∼2 h period of the aging, which has been attributed to the wall loss. However, it is not so clear whether the reduction of POA occurred due to wall loss (and/or) due to the (heterogeneous) transformations of the POA to SOA? In fact, the enhanced reduction of the olive oil POA after start of photochemical aging has been interpreted for heterogeneous oxidation of the POA, but how about the very rapid reduction of POA before the start of the aging (Figure 1)?
* * *

---

## Author Comment (AC1) · 1 Aug 2018

**General comments:**

Generally, this manuscript is well-written and contain good scientific content. I would like to recommend for publication if the authors could address some comments as the following:

**Specific comments:**

**Q1:** Line 142-144.    The author mention about the use of SMPS, but I have seen any results from these instrument. It is maybe better if the authors could show the evolution of particle number size distribution during the experiment.

R1: The evolution of particle number size distribution for the palm and olive oil experiments is shown in the following figure (now Fig. S4 in the Supplement). It is clear that the particle numbers decreased rapidly due to wall loss before the lights were switched on and a rapid growth of particles was observed after SOA formation. It should be noted that this work focused on the separation and characterization of POA and SOA from heated cooking oils, hence the evolution of particle number size distribution was not discussed in details and only provided in the Supplement now. The following sentence was added to the revised manuscript.

"Similarly, the particle numbers decreased rapidly due to wall loss before the lights were switched on and the mode particle diameters grew rapidly after SOA formation (Fig. S4)." (Line 195-197).

[Figure]

**Q2:** Smog chamber experiments: I think the author should describe in more details about the experiment: How many experiments the authors conducted for each oil cooking? The time resolution of each instruments?

R2: One experiment was conducted for each oil. The time resolution of each instrument was added to the revised manuscript.

**Q3:** What about the uncertainty of the number H:C, O:C for SOA that the author

presented in the Fig.8?

R3: According to Canagaratna et al. (2015), the uncertainty in determining O:C and H:C elemental ratios using improved-ambient method was 28% and 13%, respectively. The following sentence was added to the revised manuscript:

"The uncertainty in determining O:C and H:C ratios was 28% and 13%, respectively (Canagaratna et al., 2015)." (Line 377-378).

References:

Canagaratna, M. R., Jimenez, J. L., Kroll, J. H., Chen, Q., Kessler, S. H., Massoli, P., Hildebrandt Ruiz, L., Fortner, E., Williams, L. R., Wilson, K. R., Surratt, J. D., Donahue, N. M., Jayne, J. T., and Worsnop, D. R.: Elemental ratio measurements of organic compounds using aerosol mass spectrometry: characterization, improved calibration, and implications, Atmos. Chem. Phys., 15, 253-272, https://doi.org/10.5194/acp-15-253-2015, 2015.

---

## Author Comment (AC2) · 1 Aug 2018

Response to Reviewer #2

**General comments:**

It has been well recognized that cooking emissions substantially contribute to atmospheric organic aerosols (OA), but studies on their characterization are scarce. Hence, investigation on primary and secondary OA contributions from such emissions is highly needed. This study focused on the characterization of POA and SOA produced from heated cooking oils in laboratory experiments, based on the obtained mass spectra via PMF analysis. The residual spectrum method has also been applied to separate the POA and SOA using different tracer ion signals. The authors have successfully separated the POA and SOA produced from heated palm and olive oil by PMF analysis. Further they found that residual spectrum method is not valid to calculate POA for photochemical aging of cooking oils. The results obtained from this study are interesting and the paper is written very well. Therefore, this paper is worthy to publish in ACP, after addressing the following specific comments:

**Specific comments:**

**Q1:** It has been noted (in Section 2.1; lines 142-144) that particle number concentrations and size distributions were measured using SMPS, but did not presented the obtained data in the paper. It is important to show the particle number concentrations and size distributions during the total period of the experiments, particularly during the period before the start of photochemical aging to understand the impact of gases absorption on to seed particles on the POA loading.

R1: See response to Q1 of Reviewer #1.

**Q2:** Surprisingly, the concentrations of POA emitted from heated palm oil and olive oil were gradually reduced starting from ∼1 h and 2 h, respectively, before the start of photochemical aging and reached to almost negligible level in 4∼2 h period of the aging, which has been attributed to the wall loss. However, it is not so clear whether the reduction of POA occurred due to wall loss (and/or) due to the (heterogeneous) transformations of the POA to SOA? In fact, the enhanced reduction of the olive oil POA after start of photochemical aging has been interpreted for heterogeneous

oxidation of the POA, but how about the very rapid reduction of POA before the start of the aging (Figure 1)?

R2: In the original manuscript (Line 193-194), we have mentioned that the rapid reduction of POA before the start of the aging was due to the wall loss. Heterogeneous reactions were not expected during this period because of the absence of oxidants ($O_3$ and/or OH radicals). After the start of photochemical aging, we have clearly shown that the reduction of POA in palm oil experiment was solely due to the wall loss while both the wall loss and heterogeneous oxidation played a role in POA reduction for olive oil experiment (Line 214-231).